# Hematopoietic stem cells with granulo-monocytic differentiation state overcome venetoclax sensitivity in patients with myelodysplastic syndromes

Juan Jose Rodriguez-Sevilla[1,6], Irene Ganan-Gomez[1,6], Feiyang Ma [2], Kelly Chien [1], Monica Del Rey[3], Sanam Loghavi [4], Guillermo Montalban-Bravo [1], Vera Adema [1], Bethany Wildeman[1], Rashmi Kanagal-Shamanna [4], Alexandre Bazinet[1], Helen T. Chifotides[1], Natthakan Thongon[1], Xavier Calvo[5], Jesús María Hernández-Rivas[3], Maria Díez-Campelo [3], Guillermo Garcia-Manero [1] & Simona Colla [1] ✉

The molecular mechanisms of venetoclax-based therapy failure in patients with acute myeloid leukemia were recently clarified, but the mechanisms by which patients with myelodysplastic syndromes (MDS) acquire secondary resistance to venetoclax after an initial response remain to be elucidated. Here, we show an expansion of MDS hematopoietic stem cells (HSCs) with a granulo-monocytic-biased transcriptional differentiation state in MDS patients who initially responded to venetoclax but eventually relapsed. While MDS HSCs in an undifferentiated cellular state are sensitive to venetoclax treatment, differentiation towards a granulo-monocytic-biased transcriptional state, through the acquisition or expansion of clones with *STAG2* or *RUNX1* mutations, affects HSCs' survival dependence from BCL2-mediated anti-apoptotic pathways to TNFα-induced pro-survival NF-κB signaling and drives resistance to venetoclax-mediated cytotoxicity. Our findings reveal how hematopoietic stem and progenitor cell (HSPC) can eventually overcome therapy-induced depletion and underscore the importance of using close molecular monitoring to prevent HSPC hierarchical change in MDS patients enrolled in clinical trials of venetoclax.

The hematopoietic stem cell (HSC) hierarchy of myelodysplastic syndromes (MDS) predicts the biological mechanisms of progression after the failure of frontline hypomethylating agents (HMAs) and can guide the design or choice of second-line therapeutic approaches[1]. We

previous showed that, compared with those with a "granulocytic-monocytic progenitor (GMP) pattern" of differentiation, MDS patients with an immunophenotypic "common myeloid progenitor (CMP) pattern" of differentiation who received venetoclax-based therapy had

[1]Department of Leukemia, The University of Texas MD Anderson Cancer Center, Houston, TX, USA. [2]Department of Molecular, Cell and Developmental Biology, University of California Los Angeles, Los Angeles, CA, USA. [3]Hematology Department, University Hospital of Salamanca, IBSAL Cancer Center, Salamanca, Spain. [4]Department of Hematopathology, The University of Texas MD Anderson Cancer Center, Houston, TX, USA. [5]Laboratori de Citologia Hematològica, Servei de Patologia, Grup de Recerca Translacional en Neoplàsies Hematològiques (GRETNHE), Hospital del Mar Research Institute (IMIM), Barcelona, Spain. [6]These authors contributed equally: Juan Jose Rodriguez-Sevilla, Irene Ganan-Gomez. ✉e-mail: scolla@mdanderson.org

a shorter cumulative time to complete remission and a longer recurrence-free survival duration, primarily because venetoclax can efficiently target only "CMP pattern" HSCs, whose survival depends on BCL2.

However, MDS patients eventually failed venetoclax-based therapy after a short period of time[2].

Here, to dissect the cellular and molecular mechanisms of venetoclax-based therapy failure, we performed multi-omic analyses of sequential samples from MDS patients whose disease initially responded to venetoclax-based therapy but then relapsed.

## Results

Although further confirmed in a larger cohort of samples ($n = 28$; 12 "CMP pattern" MDS and 16 "GMP pattern" MDS) (Supplementary Fig. 1a, b and Supplementary Data 1), our survival analysis of MDS patients who were enrolled in clinical trials of venetoclax-based therapy and had longer follow-up (median time, 20.1 months) showed that those with "CMP pattern" MDS eventually lose response and/or progress to acute myeloid leukemia (AML) after an initial remission ($n = 6$ of 6 "CMP pattern" MDS patients with an initial response who did not discontinue the study) (Supplementary Data 1). These results suggest that alternative approaches are needed for these patients, who would otherwise have no other therapeutic options.

To dissect the cellular and molecular mechanisms of secondary venetoclax-based therapy failure, we performed multi-omics analyses of sequential samples from 6 "CMP pattern" MDS patients (Supplementary Data 2) whose initial disease response to venetoclax-based therapy was associated with HSC depletion (Supplementary Fig. 1c).

These analyses showed that the "CMP pattern" immunopheno-typic architecture (Supplementary Fig. 2a) and the hematopoietic stem and progenitor cell (HSPC) transcriptomic signature (Supplementary Fig. 2b, c) persisted at disease recurrence in the 3 patients with *TP53* mutations (UPN#3, UPN#4, and UPN#6), which is consistent with previous findings that *TP53* mutations confer an intrinsic resistance to BCL2 inhibition[3].

However, the HSPC hierarchy switched to "GMP pattern" MDS in the other 3 patients (UPN#1, UPN#2, and UPN#11) before venetoclax failure (Fig. 1a, b and Supplementary Fig. 3a–d). In all 3 patients, this immunophenotypic hierarchical change was associated with the acquisition or selection of clones with *STAG2* or *RUNX1* mutations, which we previously found to be enriched in "GMP pattern" MDS[1] (Fig. 1c and Supplementary Fig. 3e, f). Single-cell RNA-sequencing (scRNA-seq) analyses of mononuclear cells (MNCs) from sequential bone marrow (BM) samples from 2 of the 3 patients (Fig. 1d and Supplementary Fig. 3g) confirmed that HSCs were significantly depleted during disease remission but expanded at therapy failure (Supplementary Fig. 3h). Differential expression analyses of sequential BM samples collected during different disease stages showed that the acquisition of *STAG2*- or *RUNX1*-mutant clones not only rewired MDS HSPCs' differentiation state towards a myeloid-biased transcriptional signature (Supplementary Fig. 4a) but also changed HSCs' survival dependence from BCL2-mediated anti-apoptotic pathways to TNFα-induced pro-survival NF-κB signaling, thus enabling HSCs to evade the cytotoxic effects of venetoclax (Fig. 1e and Supplementary Fig. 4b–d).

Importantly, 3 of the 4 patients with "CMP pattern" MDS whose disease was refractory to venetoclax-based therapy (UPN#8, UPN#9, and UPN#12) carried subclones with *STAG2* and/or *RUNX1* mutations at the time of clinical trial enrollment (Supplementary Data 2). During venetoclax therapy, these clones underwent clonal evolution (Supplementary Fig. 5a), which switched the HSPC hierarchy from "CMP pattern" to "GMP pattern" MDS" (Supplementary Fig. 5b). These data confirm that *STAG2* and/or *RUNX1* mutations drive venetoclax resistance by reprogramming the HSPC architecture.

Interestingly, trisomy 8 was significantly associated with *STAG2* mutations ($P = 0.03$) and conferred a shorter duration of response to venetoclax-based therapy regardless of prior treatment in patients with "CMP pattern" MDS but not those with "GMP pattern" MDS ($n = 53$ patients treated with venetoclax-based therapies for whom immuno-phenotypic data were available) (Supplementary Fig. 6a, Supplementary Data 3). These results suggest that trisomy 8 is also a predictive biomarker of venetoclax resistance in patients with "CMP pattern" MDS.

## Discussion

The current standard of care for MDS patients is HMA therapy, which results in clinical improvements in over 50% of patients. However, the disease eventually becomes resistant to these agents. Patients with HMA-resistant MDS develop progressive cytopenias or secondary AML and have a median survival duration of only 4–6 months[4].

Venetoclax-based therapy in patients whose disease previously failed HMA therapy holds promise for improving these patients' dismal survival. However, whereas the molecular and biological mechanisms of resistance to venetoclax have recently been recently elucidated in AML[5–7], we still do not know why MDS patients whose disease failed HMA therapy acquire secondary resistance to venetoclax after an initial response[8].

Our study revealed the molecular mechanisms of venetoclax-based therapy failure in MDS. HSPCs exposed to venetoclax undergo survival pressure, which results in the acquisition or expansion of clones carrying specific genetic alterations that change these cells' dependence on BCL2-mediated pathways to NF-κB-mediated anti-apoptotic pathways for survival (Fig. 1f).

These results suggest that MDS patients receiving venetoclax-based therapy should be monitored closely for the acquisition or expansion of clones with *STAG2* or *RUNX1* mutations and enrolled in clinical trials of agents targeting NF-κB signaling effectors, such as MCL1, before their disease undergoes HSC transcriptional repro-gramming and becomes resistant to venetoclax.

## Methods

The research complies with the ethical regulations (MD Anderson Cancer Center IRB-approved human sample protocol PA15-0926).

### Human primary samples and clinical data analysis

We analyzed MDS patients who received venetoclax-based therapy at MD Anderson Cancer Center. Patients were enrolled in 1 of 3 phase I/II clinical trials (NCT04160052[2], NCT04550442[9], or NCT04655755[10]). Patient characteristics, laboratory values, and BM data, including cytogenetics and next-generation sequencing (NGS) data, were asses-sed before venetoclax-based therapy, and thereafter as clinically war-ranted. Genomic DNA was extracted from whole BM aspirates and subjected to 81-gene target polymerase chain reaction-based sequen-cing using an NGS platform as described previously[11]. Testing was performed in a Clinical Laboratory Improvement Amendments-certified laboratory. Risk stratification was performed using the Revised International Prognostic Scoring System (IPSS-R), and MDS was classified as lower-risk (IPSS-R score ≤3.5) or higher-risk (IPSS-R score >4) MDS[12,13]. Disease response was categorized according to the International Working Group 2006 criteria for MDS, and patients with responsive disease included those with complete response (CR), marrow CR (mCR), hematologic improvement (HI), or a combination of mCR and HI[14]. Response duration was defined as the time from first documented response to first documented disease progression or death, whichever occurred first. To evaluate the mechanisms of sec-ondary venetoclax-based therapy failure, we analyzed 28 MDS patients enrolled in the 3 clinical trials in whom HMA therapy had failed. To evaluate the impact of trisomy 8 on the survival of MDS patients treated with venetoclax-based therapy, we analyzed the clinical data of 53 patients who were enrolled in the 3 clinical trials regardless of prior therapies and for whom immunophenotypic data were available.

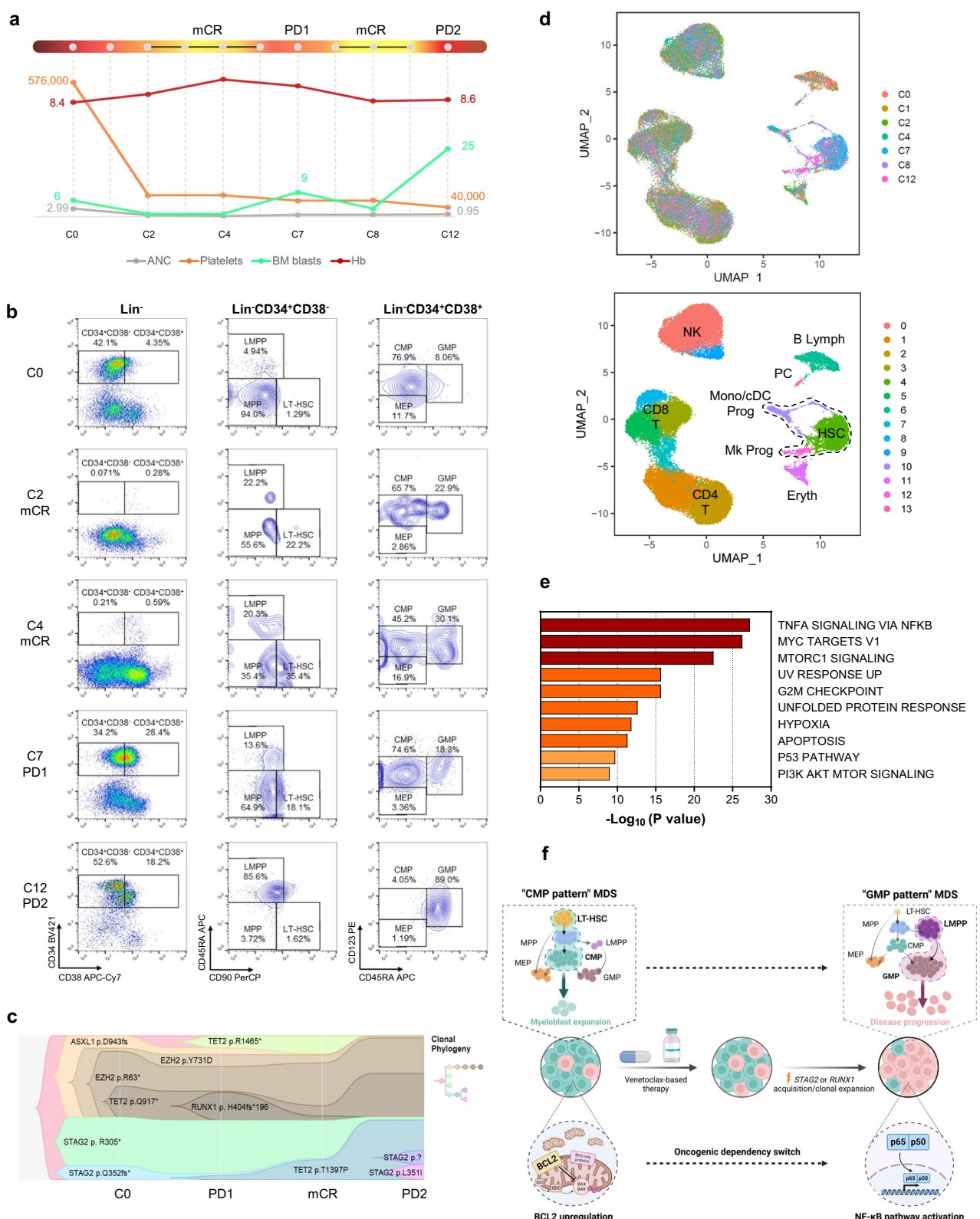

Samples were obtained in accordance with the Declaration of Helsinki from MD Anderson's Department of Leukemia under protocol PA15-0926 with the approval of the corresponding Institutional Review Boards. Written informed consent to report any information (including age, sex, and clinical parameters) was obtained from all donors, and all diagnoses were confirmed by dedicated hematopathologists. The clinical characteristics of the patients included in this study are shown in Supplementary Data 1–4. MNCs were isolated from each sample using the standard gradient separation approach with Ficoll-Paque PLUS (GE Healthcare Lifesciences, Pittsburgh, PA).

**Flow cytometry and fluorescence-activated cell sorting (FACS)**

Quantitative flow cytometric analyses and FACS of human live MNCs were performed using a previously described gating strategy and antigen panel[1,15] and antibodies against CD2 (RPA-2.10; 1:20), CD3 (SK7; 1:10), CD14 (MφP9; 1:20), CD19 (SJ25C1; 1:10), CD20 (2H7; 1:10), CD34

**Fig. 1 | Mutation-induced MDS HSCs' transcriptional reprogramming overcomes venetoclax-based therapy vulnerability. a** Schematic of UPN#1's clinical course. After HMA therapy failure (cycle 0 [C0]), UPN#1 received 5-azacitidine (75 mg/m² for 5 days) and venetoclax (400 mg/m² for 14 days) every month. The patient had mCR at cycle 2 (C2); however, after the venetoclax dose was reduced to 100 mg/m², the patient had an initial disease progression (PD1) at cycle 7 (C7). The patient had mCR after the venetoclax dose was increased to 200 mg/m² at cycle 8 (C8) but had progression to AML (PD2) at cycle 12 (C12). Hb, hemoglobin; ANC, absolute neutrophil count. Units: blasts, %; Hb, g/dL; ANC, ×10⁹/L; platelets, ×10⁹/L. **b** Flow cytometry plots of lineage (Lin)⁻CD34⁺CD38⁻ HSCs and Lin⁻CD34⁺CD38⁺ myeloid hematopoietic progenitor cells in the BM of UPN#1 at sequential timepoints before and during venetoclax-based therapy. LT-HSC long-term hematopoietic stem cells, MPP multipotent progenitors, LMPP lymphoid-primed multipotent progenitors, CMP common myeloid progenitors, GMP granulocytic-monocytic progenitors, MEP megakaryocyte erythroid progenitors. **c** Fish plot of the clonal evolution pattern inferred from NGS data for UPN#1. Phylogenetic trees show the estimated order of mutation acquisition and the proportion of subclones with different combinations of mutations at each timepoint. In UPN#1, clonal evolution was associated with the immunophenotypic HSPC hierarchical change and the acquisition of 2 *STAG2* mutations. **d** UMAP plots of scRNA-seq data from BM MNCs isolated from UPN#1 (n = 39,206). Each dot represents 1 cell. Different colors indicate sample origin (top) and cluster identity (bottom). HSC hematopoietic stem cell, Mk megakaryocytic, Mono monocytic, cDC classic dendritic, Prog progenitors, Eryth erythroblasts, NK natural killer cells, Lymph lymphocytes, PC plasma cells. Dotted lines indicate the HSPC compartment. **e** Pathway enrichment analysis of the genes that were significantly upregulated in HSCs from UPN#1 (cluster 4 in **d**) at the time of PD2 compared with those in HSCs at the time of PD1 (P adj ≤ 0.05). The top 10 Hallmark gene sets are shown. **f** Proposed working model of venetoclax-based therapy failure. After an initial response to venetoclax-based therapy, the acquisition or expansion of clones with S*TAG2* or *RUNX1* mutations reprograms the HSPC hierarchy and switches HSCs' dependence from BCL2- to NF-κB-mediated survival programs, which leads to secondary venetoclax-based therapy failure.

(581; 1:20), CD56 (B159; 1:40), CD123 (9F5; 1:20), and CD235a (HIR2; 1:40; all from BD Biosciences, Franklin Lakes, NJ); CD4 (S3.5; 1:20), CD11b (ICRF44; 1:20), CD33 (P67.6; 1:20), and CD90 (5E10; 1:10; all from Thermo Fisher Scientific, Waltham, MA); CD7 (6B7; 1:20) and CD38 (HIT2; 1:20; both from BioLegend, San Diego, CA); CD10 (SJ5-1B4; 1:20; Leinco Technologies, St. Louis, MO); and CD45RA (HI100; 1:10; Tonbo Biosciences, San Diego, CA).

FACS-purified samples were acquired with a BD Influx Cell Sorter (BD Biosciences), and the cell populations were analyzed using FlowJo software (version 10.7.1, Ashland, OR). All experiments included single-stained controls and were performed at MD Anderson's South Campus Flow Cytometry and Cellular Imaging Facility.

### scRNA-seq

scRNA-seq was performed as we described previously[1]. Briefly, FACS-purified live BM MNCs were prepared and sequenced at MD Anderson's Advanced Technology Genomics Core. Sample concentration and cell suspension viability were evaluated using a Countess II FL Automated Cell Counter (Thermo Fisher Scientific) and manual counting. Samples were normalized for input onto the Chromium Single Cell A Chip Kit (10x Genomics, Pleasanton, CA), in which single cells were lysed and barcoded for reverse-transcription. The pooled single-stranded, barcoded cDNA was amplified and fragmented for library preparation. Pooled libraries were sequenced on a NovaSeq6000 SP 100-cycle flow cell (Illumina, San Diego, CA).

The sequencing analysis was carried out using 10X Genomics' CellRanger software (version 3.0.2). Fastq files were generated using the CellRanger MkFastq pipeline (version 3.0.2). Raw reads were mapped to the human reference genome (refdata-cellranger-GRCh38-3.0.0) using the CellRanger Count pipeline. Multiple samples were aggregated using the Cellranger Aggr pipeline. The digital expression matrix was analyzed with the R package Seurat (version 3.0.2)[16] to identify different cell types and signature genes for each. Cells with fewer than 500 unique molecular identifiers or greater than 50% mitochondrial expression were removed from further analysis. The Seurat function NormalizeData was used to normalize the raw counts. Variable genes were identified using the FindVariableFeatures function. The ScaleData function was used to scale and center expression values in the dataset, and the number of unique molecular identifiers was regressed against each gene. Uniform manifold approximation and projection (UMAP) was used to reduce the dimensions of the data, and the first 2 dimensions were used in the plots. The FindClusters function was used to cluster the cells. Marker genes for each cluster were identified using the FindAllMarkers function. Cell types were annotated based on the marker genes and their match to canonical markers[17,18]. Pathway analyses of differentially expressed genes were conducted using Metascape[19]. The GMP enrichment score was calculated based on a previously validated GMP expression signature[20].

### Statistics and reproducibility

Statistical analyses were performed using R (version 4.0.320), Jamovi (version 2.0.021), and GraphPad (version 9.0.0, San Diego, CA). The 2-tailed Student t-test or Mann–Whitney test, as appropriate, and chi-square test were used to compare continuous and categorical variables, respectively. The multiple test analyses included in Supplementary Data 3 were corrected using the Bonferroni adjustment. No statistical method was used to predetermine sample size. No data were excluded from the analyses. Patient samples were selected based on of diagnosis regardless of sex and gender because MDS affect both females and males. The sex of the patients included in this study is indicated in Supplementary Data 1 and 3. Mutations with variant allele frequency values below 2% were excluded from the plot to model clonal evolution. A comprehensive summary of the mutations for UPN#1, UPN#2, UPN#11, UPN #8, UPN#9, and UPN#12 at every timepoint is provided in Supplementary Data 4. Fish plot visualization was performed using the timescape package (version 3.14) in R (version 4.2.2). The graphical abstract was made using BioRender.

### Reporting summary

Further information on research design is available in the Nature Portfolio Reporting Summary linked to this article.

## Data availability

Data sets generated in this study using scRNA-seq have been deposited at GEO under accession code GSE241417. Source data are provided as a Source data file. Source data are provided with this paper.

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

## Acknowledgements

This work was supported by philanthropic contributions to MD Anderson's AML and MDS Moon Shot Program and Support Grant CA016672, the Umberto Veronesi Foundation, and the Edward P. Evans Foundation to Dr. Simona Colla. Dr Maria Díez-Campelo was supported by the UMBRELLA Project (PI20/00970). J.J.R.-S. is a recipient of MD Anderson's Odyssey Fellowship. This work used MD Anderson's South Campus Flow Cytometry and Cellular Imaging Facility and its Advanced Technology Genomics Core, both of which are supported in part by the NIH through MD Anderson's Cancer Center Support Grant (P30 CA16672).

## Author contributions

S.C. designed the research; J.J.R.-S. guided the research; J.J.R.-S., I.G.-G., M.D.R., V.A., B.W., and N.T. performed experiments; J.J.R.-S., K.C., G.M.-B., and A.B. analyzed the clinical data; R.K.-S. and S.L. analyzed the genomic data; F.M. analyzed the scRNA-seq data; J.J.R.-S. and I.G.-G. analyzed the flow cytometry data; H.T.C. edited the manuscript. X.C., M.-D.C., J.M.H.-R., and G.G.-M. made critical intellectual contributions throughout the project; S.C. wrote the manuscript.

## Competing interests

G.G.-M. reports clinical funding from AbbVie and Amgen. All other authors report no competing interests relative to this work.
