## [Peer Review File · Nature Communications]

REVIEWER COMMENTS

Reviewer #1 (Remarks to the Author):

The manuscript by Rodriguez-Sevilla et al describes in-depth phenotypic analyses of four MDS patients undergoing treatment with venetoclax/azacytidine (ven/aza). Based on the previous outstanding work from this group, they utilize a “CMP” vs. “GMP” phenotype to identify patients with relative degrees of responsiveness to ven/aza, where CMP is more sensitive than GMP. Interesting, two of four patients bearing TP53 mutations started with a CMP phenotype and retain that profile at relapse, suggesting that disease evolution (i.e. ven/aza resistance) can be acquired while retaining the CMP profile. In contrast, two patients convert from a CMP to GMP phenotype at relapse, suggesting that disease pathogenesis is a function of conversion to the less drug-sensitive phenotype. The authors conclude that “hierarchical rewiring” of HSCs occurs to drive the conversion from a CMP to GMP phenotype, and also note that acquisition or expansion of clones bearing trisomy 8 and either STAG2 or RUNX1 mutations is associated with the conversion to the GMP profile. Based on these findings, the authors propose that molecular monitoring should be incorporated into ven/aza MDS studies as a means to prevent HSC transcriptional programming before the disease becomes resistant to therapy.

This is a fascinating study and continues the excellent work of this group in probing MDS pathogenesis. The study would be strengthened though by addressing the following points:

Major points:

- 1) The interpretation of the data is that reprogramming/rewiring occurs to drive the conversion from CMP to GMP. While this may be true, it seems an equally plausible explanation is that GMP phenotype HSCs are already present at a low level in these patients, and that their emergence is simply a process of selection. Consequently, it seems premature to conclude that the phenomenon described must be reprogramming/rewiring. It is suggested that the text be modified to consider selection as a possibility as well.
- 2) The limited number of patients analyzed here is a concern. Based on their Nature Medicine paper from last year, the data seem to imply that the phenotypes of MDS are a binary choice, either CMP or GMP. In the four patients analyzed, two retain CMP and two convert to GMP, but trying to associate that behavior with underlying mutations with so few patients seem a bit tenuous. Increasing the numbers of specimens analyzed would greatly strengthen the conclusions.

Minor point:

The authors seem to argue that current studies should perform molecular monitoring as a means to modify therapy should a GMP phenotype begin to emerge. While this reviewer certainly agrees that such monitoring could have great value in the future, there is currently not a compelling case to conduct such an analysis for patient benefit, since there is no current intervention to provide for patients converting to a GMP phenotype. The value of the study, which is considerably, is the fascinating insight into MDS pathogenesis, which clearly highlights the need to find therapies that better target the GMP phenotype. There is no need to claim that the approach has immediate diagnostic value.

Reviewer #2 (Remarks to the Author):

Venetoclax, the first-in-class BCL-2 inhibitor, has proven efficacy and safety in combination with hypomethylating agents for treatment of high-risk myeloid diseases such as Acute Myeloid Leukemia (AML) and high-risk Myelodysplastic Syndromes (HR-MDS). While molecular mechanisms of venetoclax resistance have been recently unraveled in AML, mechanisms by which MDS patients acquire resistance to venetoclax-based therapies remain to be elucidated.

In the present manuscript Rodriguez-Sevilla and coworkers demonstrate that MDS hematopoietic stem cells (HSC) undergo a hierarchical differentiation reprogramming toward a granulo-monocytic-progenitor (GMP) transcriptional state which changes HSCs' survival dependence from BCL-2-mediated anti-apoptotic pathways to TNF α -induced pro-survival NF- κ B signaling and thus overcomes venetoclax-cytotoxic effects. The GMP-pattern of differentiation was prominent in clones with an isolated trisomy 8 and STAG2 or RUNX1 mutations, thus proposing these (cyto)genetic aberrations as potential markers for venetoclax resistance.

Overall, this is a well-written comprehensive manuscript presenting novel findings regarding venetoclax resistance mechanisms in MDS with potential for clinical translation. However, the number of patients analyzed in this study is quite low, the study is mainly descriptive at its current state and mechanistic data validating the findings from scRNA-seq analysis are lacking.

Main points:

1. The patient cohort analyzed by scRNA-seq is quite small comprising only 4 patient samples post-VEN. Two of the analyzed patients carry TP53 mutations known to mediate VEN resistance as reported in AML. Thus, only for 2 patients the switch from CMP- to GMP-pattern and acquisition of trisomy 8 clones, STAG2 and RUNX1 mutations at therapy failure are identified. This is clearly not enough to convincingly draw the conclusion that trisomy 8 and STAG2/ RUNX1 mutations are biomarkers for VEN failure. Accordingly, more CMP-pattern patients with secondary VEN-resistance should be analyzed to validate the initial results.

2. scRNA-seq data are not validated by independent assays, such as overexpression of identified STAG2/RUNX1 mutations in cell lines, PDX samples or primary samples with subsequent analysis of VEN response. Such experiments are highly needed to proof the hypothesis that identified mutations are biomarkers for VEN-resistance in MDS. Further, it should be demonstrated whether acquisition of respective STAG2/RUNX1 mutations alters expression of BCL2 and NF- κ B target genes.

3. Suppl. Fig.1 n: The authors identified that genes involved in TNF α signaling through the NF- κ B pathway were significantly upregulated in HSCs at VEN failure. Among these, MCL1 known to mediate VEN-resistance in AML, is upregulated. Thus, shifting HSC's survival dependence from BCL-2 to MCL-1 could be a mechanism of VEN-resistance in MDS as well, which should be experimentally tested.

Minor points:

1. Suppl. Fig. 1c- Fish plot of clonal evolution for UPN#1: Before VEN treatment and at subsequent treatment cycles clones with STAG2 R305* and STAG2 Q352fs mutations were present, whereas at secondary resistance (PD2) STAG2 L351I mutant clones grew out. Is it known whether the different STAG2 mutations all induce LOF of STAG2? As STAG2 mutations correlate with MDS response to hypomethylating agents and the analyzed patients have been treated with VEN + AZA (or DAC, respectively) in clinical trials, does acquisition of STAG2 mutations rather reflect secondary resistance to HMAs in the combinatorial treatment?

Further, in a previous publication (PMID: 35241842) the authors showed that GMP- pattern MDS is associated with an increased frequency of STAG2, RUNX1, DNMT3A and BCOR mutations (in the absence of VEN- therapeutic pressure). Thus, are STAG2 and RUNX1 mutations rather „biomarkers“ for GMP-pattern MDS (and distinct transcriptional states) than specific biomarkers for VEN-resistance?

2. Suppl. Fig. 1d- Flow Cytometry Plots of Lin- CD34+/CD38- HSCs and CD34+/CD38+ Progenitors

of two CMP pattern patients with TP53 mutation: For plot „C5 Relapse“ percentage of CD34+/CD38- HSCs and CD34+/CD38+ progenitors is below 2% (and thus lower as at mCR). How was relapse defined here?

RESPONSE TO REVIEWERS' COMMENTS

Reviewer #1 (Remarks to the Author):

The manuscript by Rodriguez-Sevilla et al describes in-depth phenotypic analyses of four MDS patients undergoing treatment with venetoclax/azacytidine (ven/aza). Based on the previous outstanding work from this group, they utilize a “CMP” vs. “GMP” phenotype to identify patients with relative degrees of responsiveness to ven/aza, where CMP is more sensitive than GMP. Interesting, two of four patients bearing TP53 mutations started with a CMP phenotype and retain that profile at relapse, suggesting that disease evolution (i.e. ven/aza resistance) can be acquired while retaining the CMP profile. In contrast, two patients convert from a CMP to GMP phenotype at relapse, suggesting that disease pathogenesis is a function of conversion to the less drug-sensitive phenotype. The authors conclude that “hierarchical rewiring” of HSCs occurs to drive the conversion from a CMP to GMP phenotype, and also note that acquisition or expansion of clones bearing trisomy 8 and either STAG2 or RUNX1 mutations is associated with the conversion to the GMP profile. Based on these findings, the authors propose that molecular monitoring should be incorporated into ven/aza MDS studies as a means to prevent HSC transcriptional programming before the disease becomes resistant to therapy. This is a fascinating study and continues the excellent work of this group in probing MDS pathogenesis. The study would be strengthened though by addressing the following points:

We thank the reviewer for the positive comments. We hope that our explanations below and the inclusion of additional patients in the study, which was done in response to the reviewer’s questions, will alleviate the reviewer’s concerns.

Major points:

1) The interpretation of the data is that reprogramming/rewiring occurs to drive the conversion from CMP to GMP. While this may be true, it seems an equally plausible explanation is that GMP phenotype HSCs are already present at a low level in these patients, and that their emergence is simply a process of selection. Consequently, it seems premature to conclude that the phenomenon described must be reprogramming/rewiring. It is suggested that the text be modified to consider selection as a possibility as well.

We agree with the reviewer. The acquisition or expansion of clones carrying *STAG2* or *RUNX1* mutations drives venetoclax resistance. These mutations are associated with a switch in the hematopoietic hierarchy from “CMP pattern” MDS to “GMP pattern” MDS. This switch could be induced either by HSPC reprogramming towards a more differentiated myeloid state (likely in the case of a newly acquired mutation, e.g., in UPN#2) or by the clonal selection of a preexisting subclone (likely in the case of the evolution of subclones with *STAG2* mutations, e.g., UPN#2 and UPN#11). Following the reviewer’s suggestions, we have changed the text in the manuscript. The newly added parts are highlighted in red in the text.

2) The limited number of patients analyzed here is a concern. Based on their Nature Medicine paper from last year, the data seem to imply that the phenotypes of MDS are a binary choice, either CMP or GMP. In the four patients analyzed, two retain CMP and two

convert to GMP, but trying to associate that behavior with underlying mutations with so few patients seem a bit tenuous.

The revised manuscript includes data from 2 additional patients with “CMP pattern” MDS in whom venetoclax therapy failed after an initial response (UPN#6 and UPN#11). We also analyzed an additional 4 patients (UPN#7, UPN#8, UPN#9, and UPN#12) whose disease never responded to venetoclax (revised Supplementary Table 2). All results are summarized below and included in the manuscript.

Among the 28 patients who had HMA therapy failure and enrolled in clinical trials of venetoclax-based therapy, 12 were classified as having “CMP pattern” MDS. Two patients dropped out of the trial: one experienced venetoclax-related toxicities after an initial response, and one underwent allogeneic stem cell transplantation after an initial response (Supplementary Table 1). Of the remaining 10 patients, 6 had responses to venetoclax-based therapies but later experienced disease relapse or progression (revised Supplementary Table 2). Of these 6 patients, 3 (UPN#1, UPN#2, and UPN#11) had acquired or expanded clones carrying *STAG2* or *RUNX1* mutations at the time of therapy failure, and these patients’ hematopoietic stem cell hierarchy switched from “CMP pattern” to “GMP pattern” MDS. We have included the results of the new analyses of UPN#11 in Supplementary Figures 1i,j,l.

The other 3 patients had multiple clones carrying *TP53* mutations. In 2 of these patients (UPN#4 and UPN#6), *TP53* clones further expanded at therapy failure, as shown by the VAFs of *TP53* mutations (revised Supplementary Table 2). In one of the 3 patients (UPN#3), the *TP53* mutation VAF was already close to 50% at the time of enrollment in the trial (Supplementary Table 2). In all 3 patients with *TP53* mutations, the HSPC hierarchy never changed during venetoclax treatment. We have included the results of the flow cytometry analysis of UPN#6 (not previously analyzed) in Supplementary Figure 1d.

To further support our conclusions, we analyzed the 4 patients with “CMP pattern” MDS whose disease never responded to venetoclax-based therapy (UPN#7, UPN#8, UPN#9, and UPN#12; revised Supplementary Table 2). At the time of their enrollment in the clinical trials of venetoclax-based therapy, 3 of the 4 patients (UPN#8, UPN#9, and UPN#12) had clones carrying *STAG2* or *RUNX1* mutations, which dramatically expanded during venetoclax treatment (new Supplementary Figure 1s and revised Supplementary Table 2). In the 2 patients with available sequential samples, the expansion of clones with *STAG2* or *RUNX1* mutations was associated with a switch in the cellular architecture from “CMP pattern” MDS to “GMP pattern” MDS (new Supplementary Figure 1t). One patient (UPN#7) had 3 different mutations in *TP53*, 2 of which (possibly in the same clone) expanded after treatment (revised Supplementary Table 2).

Together, these data confirm that: 1) mutations in *STAG2* or *RUNX1* drive venetoclax failure by reprogramming or expanding specific HSPC hierarchies (this is a novel and important finding); and 2) *TP53* mutations confer resistance to venetoclax in MDS patients, as has been reported in AML patients (*Kim et al., Cancer 2021*).

Although we have a limited cohort of samples, the results are very consistent and, in our modest opinion, sufficient to support our conclusions.

Minor point:

The authors seem to argue that current studies should perform molecular monitoring as a means to modify therapy should a GMP phenotype begin to emerge. While this reviewer certainly agrees that such monitoring could have great value in the future, there is currently not a compelling case to conduct such an analysis for patient benefit, since there is no current intervention to provide for patients converting to a GMP phenotype. The value of the study, which is considerably, is the fascinating insight into MDS

pathogenesis, which clearly highlights the need to find therapies that better target the GMP phenotype. There is no need to claim that the approach has immediate diagnostic value.

We agree with the reviewer that molecular monitoring may not have an immediate clinical impact in patients in whom venetoclax therapy has failed, given the current absence of approved therapies for patients whose disease progresses after an initial response. However, an ongoing clinical trial of the MCL1 inhibitor AMG-176 at MD Anderson (NCT05209152) is enrolling patients in whom HMA therapy has failed independently of previous venetoclax exposure. This trial was supported by our preclinical studies' in vitro findings showing that: 1)

MCL1 expression was significantly upregulated in HSPCs at the time of venetoclax failure (**A, on the left**). 2) AMG-176 alone (at a dose that does not affect HSCs from healthy donors; **B, on the left**) or in combination with venetoclax significantly depletes CD34⁺CD38⁻ HSCs from patients in whom

venetoclax-based therapy failed and who had a "GMP pattern" MDS architecture at the time of failure (**C, above**). That being said, we have revised the abstract and the conclusion to reflect the reviewer's perspective.

Reviewer #2 (Remarks to the Author):

Venetoclax, the first-in-class BCL-2 inhibitor, has proven efficacy and safety in combination with hypomethylating agents for treatment of high-risk myeloid diseases such as Acute Myeloid Leukemia (AML) and high-risk Myelodysplastic Syndromes (HR-MDS). While molecular mechanisms of venetoclax resistance have been recently unraveled in AML, mechanisms by which MDS patients acquire resistance to venetoclax-based therapies remain to be elucidated. In the present manuscript Rodriguez-Sevilla and coworkers demonstrate that MDS hematopoietic stem cells (HSC) undergo a hierarchical differentiation reprogramming toward a granulo-monocytic-progenitor (GMP) transcriptional state which changes HSCs' survival dependence from BCL-2-mediated anti-apoptotic pathways to TNF α -induced pro-survival NF- κ B signaling and thus overcomes venetoclax- cytotoxic effects. The GMP-pattern of differentiation was prominent in clones with an isolated trisomy 8 and STAG2 or RUNX1 mutations, thus proposing these (cyto)genetic aberrations as potential markers for venetoclax resistance. Overall, this is a well-written comprehensive manuscript presenting novel findings regarding venetoclax resistance mechanisms in MDS with potential for clinical translation. However, the number of patients analyzed in this study is quite low, the study is mainly descriptive at its current state and mechanistic data validating the findings from scRNA-seq analysis are lacking.

We thank the reviewer for the positive criticism. We hope that our explanations below and the results of our additional experiments performed in response to the reviewer's questions will alleviate the reviewer's concerns.

Main points:

1. The patient cohort analyzed by scRNA-seq is quite small comprising only 4 patient samples post-VEN. Two of the analyzed patients carry TP53 mutations known to mediate VEN resistance as reported in AML. Thus, only for 2 patients the switch from CMP- to GMP-pattern and acquisition of STAG2 and RUNX1 mutations at therapy failure are identified. This is clearly not enough to convincingly draw the conclusion that STAG2/RUNX1 mutations are biomarkers for VEN failure. Accordingly, more CMP-pattern patients with secondary VEN-resistance should be analyzed to validate the initial results.

The revised manuscript includes data from 2 additional patients with "CMP pattern" MDS in whom venetoclax therapy failed after an initial response (UPN#6 and UPN#11). We also analyzed an additional 4 patients (UPN#7, UPN#8, UPN#9, and UPN#12) whose disease never responded to venetoclax (revised Supplementary Table 2). All results are summarized below and included in the manuscript.

Among the 28 patients who had HMA therapy failure and enrolled in clinical trials of venetoclax-based therapy, 12 were classified as having "CMP pattern" MDS. Two patients dropped out of the trial: one experienced venetoclax-related toxicities after an initial response, and one underwent allogeneic stem cell transplantation after an initial response (Supplementary Table 1). Of the remaining 10 patients, 6 had responses to venetoclax-based therapies but later experienced disease relapse or progression (revised Supplementary Table 2). Of these 6 patients, 3 (UPN#1, UPN#2, and UPN#11) had acquired or expanded clones carrying *STAG2* or *RUNX1* mutations at the time of therapy failure, and these patients' hematopoietic stem cell hierarchy switched from "CMP pattern" to "GMP pattern" MDS. We have included the results of the new analyses of UPN#11 in Supplementary Figures 1i,j,l.

The other 3 patients had multiple clones carrying *TP53* mutations. In 2 of these patients (UPN#4 and UPN#6), *TP53* clones further expanded at therapy failure, as shown by the VAFs of *TP53* mutations (revised Supplementary Table 2). In one of the 3 patients (UPN#3), the *TP53* mutation VAF was already close to 50% at the time of enrollment in the trial (Supplementary Table 2). In all 3 patients with *TP53* mutations, the HSPC hierarchy never changed during venetoclax treatment. We have included the results of the flow cytometry analysis of UPN#6 (not previously analyzed) in Supplementary Figure 1d.

To further support our conclusions, we analyzed the 4 patients with “CMP pattern” MDS whose disease never responded to venetoclax-based therapy (UPN#7, UPN#8, UPN#9, and UPN#12; revised Supplementary Table 2). At the time of their enrollment in the clinical trials of venetoclax-based therapy, 3 of the 4 patients (UPN#8, UPN#9, and UPN#12) had clones carrying *STAG2* or *RUNX1* mutations, which dramatically expanded during venetoclax treatment (new Supplementary Figure 1s and revised Supplementary Table 2). In the 2 patients with available sequential samples, the expansion of clones with *STAG2* or *RUNX1* mutations was associated with a switch in the cellular architecture from “CMP pattern” MDS to “GMP pattern” MDS (new Supplementary Figure 1t). One patient (UPN#7) had 3 different mutations in *TP53*, 2 of which (possibly in the same clone) expanded after treatment (revised Supplementary Table 2).

Together, these data confirm that: 1) mutations in *STAG2* or *RUNX1* drive venetoclax failure by reprogramming or expanding specific HSPC hierarchies (this is a novel and important finding); and 2) *TP53* mutations confer resistance to venetoclax in MDS patients, as has been reported in AML patients (*Kim et al., Cancer 2021*).

Although we have a limited cohort of samples, the results are very consistent and, in our modest opinion, sufficient to support our conclusions.

2. scRNA-seq data are not validated by independent assays, such as overexpression of identified *STAG2*/*RUNX1* mutations in cell lines, PDX samples or primary samples with subsequent analysis of VEN response. Such experiments are highly needed to proof the hypothesis that identified mutations are biomarkers for VEN-resistance in MDS. Further, it should be demonstrated whether acquisition of respective *STAG2*/*RUNX1* mutations

alters expression of *BCL2* and *NF-κB* target genes.

Our data do not suggest that mutations in *RUNX1* or *STAG2* directly confer resistance to venetoclax. Instead, we propose that these mutations affect hematopoiesis, induce myeloid differentiation of

LT-HSCs, and increase the self-renewal capabilities of LMPPs and GMPs, as previously shown in conditional mouse models of *RUNX1* or *STAG2* deletion (Behrens et al., *Blood* 2016; Viny et al., *Cell Stem Cell* 2019). We have already demonstrated that MDS LMPPs and GMPs rely on NF- κ B-mediated pathways, but not BCL2-mediated pathways, to maintain survival under therapy (Ganan-Gomez et al., *Nature Medicine* 2022). Our data suggest that the differentiation state of the cells, rather than the direct molecular effects of the mutations, induces resistance to venetoclax.

To further support our conclusions, we used the SKM-1 cell line, previously established from a patient with MDS whose disease progressed to myelomonocytic leukemia. This cell line, which carries mutations affecting *STAG2*, *ASXL1*, *KRAS*, and *TP53*, faithfully recapitulates the molecular complexity of MDS patients in whom HMA and venetoclax-based therapies have failed. Given that the *STAG2* mutation R259* is a truncating mutation, we restored *STAG2* expression through lentiviral gene delivery (A, above). Consistent with the hypothesis that the differentiation state of the cells, rather than the direct molecular effects of the mutations, induces resistance to venetoclax, the *STAG2*-re-expressing SKM-1 cells did not show any difference in the expression of BCL2 or phospho-p65 (the effector of NF- κ B pathway activation) (B, above) or in sensitivity to venetoclax alone or in combination with HMA therapy (C, above).

3. Suppl. Fig.1n: The authors identified that genes involved in TNF α signaling through the NF- κ B pathway were significantly upregulated in HSCs at VEN failure. Among these, MCL1 known to mediate VEN-resistance in AML, is upregulated. Thus, shifting HSC's survival dependence from BCL-2 to MCL-1 could be a mechanism of VEN-resistance in MDS as well, which should be experimentally tested.

The reviewer is correct. MCL1 expression was significantly upregulated in HSPCs at the time of venetoclax failure (A, on the left). Following the reviewer's suggestions, to understand whether targeting MCL1 can overcome resistance to venetoclax therapy in "GMP pattern" MDS after venetoclax

failure, we treated CD34⁺CD38⁻ HSCs isolated from the BM of a patient with "GMP pattern" MDS after venetoclax failure with the MCL1 inhibitor AMG-176 alone or in combination with venetoclax. AMG-176 alone or in combination with venetoclax, at a dose that did not affect

CD34⁺CD38⁻ cells from healthy donors (20 nM; **B above**) significantly reduced CD34⁺CD38⁻ cells (**C, above**). As correctly hypothesized by the reviewer, these findings suggest that MCL-1 upregulation is a mechanism of resistance to venetoclax and that “GMP-pattern” MDS patients in whom venetoclax-based therapy has failed may benefit from therapy based on MCL1 inhibitors.

Minor points:

1.Suppl. Fig. 1c- Fish plot of clonal evolution for UPN#1: Before VEN treatment and at subsequent treatment cycles clones with STAG2 R305* and STAG2 Q352fs mutations were present, whereas at secondary resistance (PD2) STAG2 L351I mutant clones grew out. Is it known whether the different STAG2 mutations all induce LOF of STAG2?

Disease relapse in UPN#1 was driven by the expansion of the *STAG2* Q352fs* clone, which outcompeted the original *STAG2* R305* clone. The *STAG2* L351I mutant clone was only subclonal at the time of therapy relapse, and it did not drive disease relapse (**Figure below**). That being said, no functional validation studies have tested the effect of these mutations on STAG2 function. *STAG2* R305* and *STAG2* Q352fs mutations are truncating and frameshift mutations, respectively. *STAG2* L351I is a missense mutation. All these mutations are inside the cohesion domain (spanning amino acids 102-395 of the protein). Mutations affecting this domain are expected to be loss-of-function mutations.

Independently of clarifying which of the 3 mutations most strongly affects STAG2 function (which is beyond the scope of this study), these data suggest that venetoclax therapy induces a

selective pressure on HSPCs that results in the new acquisition or expansion of clones with mutations affecting *STAG2* or *RUNX1*. These mutations switch HSPCs' hierarchical architecture and their survival

dependencies, as extensively discussed in response to the reviewers' questions above.

As STAG2 mutations correlate with MDS response to hypomethylating agents and the analyzed patients have been treated with VEN + AZA (or DAC, respectively) in clinical trials, does acquisition of STAG2 mutations rather reflect secondary resistance to HMAs in the combinatorial treatment?

Only one study showed that patients with *STAG2* mutations have better outcomes under HMA therapy (Thota et al., Blood 2014). However, the findings of that correlative study have never been confirmed (e.g., Montalban-Bravo et al., Oncotarget 2018). On the contrary, our previous

study showed that *STAG2* mutations were acquired at the time of progression after HMA therapy failure (Ganan-Gomez et al, *Nature Medicine* 2022). In the present study, clones carrying *STAG2* mutations expanded during venetoclax-based therapy in 2 patients (UPN#1 and UPN#11). We have no information about the *STAG2* VAF during HMA therapy in UPN#1. However, in UPN#11, clones with *STAG2* mutations did not expand during HMA therapy and did not drive disease relapse (*STAG2* VAF at baseline: 12%; *STAG2* VAF at HMA therapy failure: 14%) but expanded only after venetoclax-based therapy (*STAG2* VAF at venetoclax therapy failure: 80%). Overall, these data demonstrate that *STAG2* mutations promote resistance to venetoclax and not to HMA therapy.

Further, in a previous publication (PMID: 35241842) the authors showed that GMP-pattern MDS is associated with an increased frequency of *STAG2*, *RUNX1*, *DNMT3A* and *BCOR* mutations (in the absence of VEN- therapeutic pressure). Thus, are *STAG2* and *RUNX1* mutations rather, biomarkers for GMP-pattern MDS (and distinct transcriptional states) than specific biomarkers for VEN-resistance?

The reviewer is correct. *STAG2* and *RUNX1* mutations are associated with “GMP pattern” MDS when they are in the driver mutant clone. In the present study, we showed that venetoclax resistance is driven by the rewiring of the HSC differentiation trajectory induced by the acquisition and expansion of these mutant clones, rather than the mutations themselves. Consistent with our hypothesis, studies in conditional mouse models of *RUNX1* and *STAG2* deletion demonstrated that *RUNX1* or *STAG2* depletion induces the increased self-renewal capability and expansion of LMPPs and GMPs (Behrens et al., *Blood* 2016; Viny et al., *Cell Stem Cell* 2019).

2. Suppl. Fig. 1d- Flow Cytometry Plots of Lin- CD34+/CD38- HSCs and CD34+/CD38+ Progenitors of two CMP pattern patients with TP53 mutation: For plot C5 relapse percentage of CD34+/CD38- HSCs and CD34+/CD38+ progenitors is below 2% (and thus lower as at mCR). How was relapse defined here?

The reviewer refers to patient UPN#4 in Supplementary Table 2. Relapse was defined

according to the IWG 2006 criteria (Cheson et al., *Blood* 2006). These criteria are based on BM blast percentages and transfusion dependency. The parameters included in the **Table and Figure above** clearly demonstrate the failure of venetoclax-based therapy in this patient.

REVIEWERS' COMMENTS

Reviewer #1 (Remarks to the Author):

The inclusion of additional patients is very helpful and substantially strengthens the manuscript. The reviewer appreciates seeing the data on use of an MCL inhibitor for HMA failure patients, hopefully this will be a good option for those patients.

The reviewer also appreciates the text modifications to include expansion of pre-existing clones as a possible mechanism to explain the CMP to GMP switching. However, it is suggested that some additional editing would improve clarity. The manuscript still uses the term "HSC differentiation state reprogramming" in the title and concludes that "HSC plasticity" (in the abstract and last paragraph) is the underlying mechanism. This is inherently confusing and undermines the value of the paper. If the mechanism is indeed expansion of a pre-existing clone, then by definition that is not HSC "plasticity". While this may be largely a semantic issue, in context, the term plasticity is generally not used to describe the outgrowth of a pre-existing clone. Further, the authors present no direct data supporting plasticity as the main mechanism. The manuscript simply needs to reflect that either HSC plasticity or clonal expansion are the two possible mechanisms.

Reviewer #2 (Remarks to the Author):

In the original version of their manuscript Rodriguez-Sevilla and coworkers demonstrate that MDS hematopoietic stem cells (HSC) undergo a hierarchical differentiation reprogramming toward a granulomonocytic-progenitor (GMP) transcriptional state which changes HSCs' survival dependence from BCL-2-mediated anti-apoptotic pathways to TNF α -induced pro-survival NF- κ B signaling and thus overcomes venetoclax- cytotoxic effects. The GMP-pattern of differentiation was prominent in clones with an isolated trisomy 8 and STAG2 or RUNX1 mutations, thus proposing these (cyto)genetic aberrations as potential markers for venetoclax resistance.

Although results obtained from this study were highly interesting, several issues remained in the original version. The current revised version of the manuscript now addresses the reviewer's suggestions and clarifies all open questions and issues. Specifically, inclusion of scRNA-seq data from additional patients clearly improves the validity of the story.

RESPONSE TO REVIEWERS' COMMENTS

Reviewer #1

The inclusion of additional patients is very helpful and substantially strengthens the manuscript. The reviewer appreciates seeing the data on use of an MCL inhibitor for HMA failure patients, hopefully this will be a good option for those patients.

The reviewer also appreciates the text modifications to include expansion of pre-existing clones as a possible mechanism to explain the CMP to GMP switching. However, it is suggested that some additional editing would improve clarity. The manuscript still uses the term "HSC differentiation state reprogramming" in the title and concludes that "HSC plasticity" (in the abstract and last paragraph) is the underlying mechanism. This is inherently confusing and undermines the value of the paper. If the mechanism is indeed expansion of a pre-existing clone, then by definition that is not HSC "plasticity". While this may be largely a semantic issue, in context, the term plasticity is generally not used to describe the outgrowth of a pre-existing clone. Further, the authors present no direct data supporting plasticity as the main mechanism. The manuscript simply needs to reflect that either HSC plasticity or clonal expansion are the two possible mechanisms.

Based on the reviewer's suggestion we have modified the title of the manuscript and eliminated the word "plasticity" from the abstract.